# Bag of Tricks for FGSM Adversarial Training

## Abstract

Adversarial training (AT) with samples generated by Fast Gradient Sign Method (FGSM), also known as FGSM-AT, is a computationally simple method to train robust networks. However, during its training procedure, an unstable mode of "catastrophic overfitting" has been identified in Wong et al. [2020], where the robust accuracy abruptly drops to zero within a single training step. Existing methods use gradient regularizers or random initialization tricks to attenuate this issue, whereas they either take high computational cost or lead to lower robust accuracy. In this work, we provide the first study which thoroughly examines a collection of tricks from three perspectives: *Data Initialization, Network Structure, and Optimization*, to overcome the catastrophic overfitting in FGSM-AT. Surprisingly, we find that simple tricks, *i.e.*, masking partial pixels (even without randomness), setting a large convolution stride and smooth activation functions, or regularizing the weights of the first convolutional layer can effectively tackle the overfitting issue. Extensive results on a range of network architectures validate the effectiveness of each proposed tricks, and the combinations of tricks are also investigated. For example, trained with PreActResNet-18 on CIFAR-10, our method attains 51.3% accuracy against PGD-10 attacker and 46.4% accuracy against AutoAttack, demonstrating that pure FGSM-AT is capable of enabling robust learners. We will release our code to encourage future exploration on unleashing the potential of FGSM-AT.

## 1 Introduction

Convolution neural networks (CNNs), though achieving compelling performances on various visual recognition tasks, are vulnerable to adversarial perturbations Szegedy et al. [2013]. To effectively defend against such malicious attacks, adversarial examples are utilized as training data for enhancing model robustness, a process known as adversarial training (AT). To generate adversarial examples, one of the leading approaches is to perturb the data using the sign of the image gradients, namely the Fast Gradient Sign Method (FGSM) Goodfellow et al. [2015].

The adversarial training with FGSM (FGSM-AT) is computationally efficient, and it lies the foundation for many followups Kurakin et al. [2016], Madry et al. [2018], Zhang et al. [2019]. Nonetheless, interestingly, FGSM-AT is not widely used today because of the catastrophic overfitting: the model robustness will collapse after a few training epochs Wong et al. [2020]. To mitigate the catastrophic overfitting and stabilize FGSM-AT, several methods have been proposed. For instance, Wong et al. [2020] pre-add uniformly random noises to images to generate adversarial examples, *i.e.*, turn the FGSM attacker into the PGD-1 attacker. Andriushchenko and Flammarion [2020] propose GradAlign, which regularizes the AT via maximizing the gradient alignment of the perturbations. While these approaches successfully alleviate the catastrophic overfitting, some limitations . For

Submitted to 35th Conference on Neural Information Processing Systems (NeurIPS 2021). Do not distribute.

example, GradAlign requires an extra forward pass compared to the vanilla FGSM-AT, which significantly increases the computational cost; Fast-AT in Wong et al. [2020] shows a relatively lower robustness, and may still collapse if training with larger networks.

In this paper, we aim to develop more effective and computationally efficient solutions for attenuating this catastrophic overfitting. Specifically, we revisit FGSM-AT and propose to stabilize its training from the following three perspectives:

- **Data Initialization.** Following the idea of adding random perturbations Madry et al. [2018], Wong et al. [2020], we propose to randomly mask a subset of the input pixels to stabilize FGSM-AT, dubbed FGSM-Mask. Surprisingly, additional analysis suggests that the masking process does not necessarily need to be set as random during training—we find that applying a pre-defined masking pattern to the training set also effectively stabilizes FGSM-AT. This observation also holds for adding perturbations as the initialization in Wong et al. [2020], challenging the general belief that randomness is the key factor for stabilizing AT.

- **Network Structure.** We identify two architectural elements that affect FGSM-AT. Firstly, in addition to boosting robustness as shown in Xie et al. [2020], we find a smoother activation function can make FGSM-AT more stable. Secondly, we find vanilla FGSM-AT can effectively train ViTs without showing catastrophic overfitting. We conjecture this phenomenon may be related to how CNNs and ViTs extract features: *i.e.*, CNNs extract features from overlapped image regions (where stride size < kernel size), while ViT extract features from non-overlapped image patches (where stride size = kernel size). By simply increasing the stride size of the first convolution layer in a CNN, we find the resulted model can stably train with FGSM-AT.

- **Optimization.** GradAlign Andriushchenko and Flammarion [2020] stabilizes the FGSM-AT by setting the norm of the gradients as a regularization term. To further reduce the computational cost, we propose ConvNorm, a regularization term that simply constrains the weights of the first convolution layer. Different from GradAlign which introduces a significant amount of extra computations, our ConvNorm can work as nearly computationally efficient as the vanilla FGSM-AT.

**Our contributions.** In summary, we discover a bag of tricks that effectively alleviate the catastrophic overfitting in FGSM-AT from different perspectives. We extensively validate the effectiveness of our methods with a range of different network structures on the popular CIFAR-10 dataset. Based on our results, we can conclude that the pure FGSM-AT is capable of enabling robust learners.

## 2 Preliminaries

Given a neural classifier $f$ with parameters $\theta$, we denote $x$ and $y$ as input data and labels from the data generator $D$, respectively . $\delta$ represents the perturbations and $\mathcal{L}$ is the cross-entropy loss typically used for image classification tasks.

**Adversarial Training:** We can formulate the adversarial training as an optimization problem Madry et al. [2018] as:

$$\min_{\theta} \mathbb{E}_{(x,y) \sim D} \big[ \max_{\delta \in \Delta} \mathcal{L}(f_\theta(x + \delta), y) \big]. \tag{1}$$

Among different methods for generating adversarial examples, we chose two popular ones to study:

- **FGSM:** Goodfellow et al. [2015] first propose Fast Gradient Sign Method (FGSM) to generate the perturbation $\delta$ as follows:
$$\delta = \epsilon \, \text{sign}(\nabla_x \mathcal{L}(f_\theta(x), y)), \tag{2}$$

- **PGD:** Madry et al. [2018] propose a strong iterative version with a random start based on FGSM, name projected gradient descent (PGD) as:
$$x_{t+1} = \Pi_{\|\delta\|_\infty \leq \epsilon} \left( x_t + \alpha \text{sign}(\nabla_{x_t} \mathcal{L}(f_\theta(x_t), y)) \right), \tag{3}$$

where the $\alpha$ denotes the step size of each iteration. PGD provides a better choice for adversarial examples, but it will also cost much more time than FGSM. In the following sections, we call adversarial training with FGSM as FGSM-AT and correspondingly, PGD-AT. where $\epsilon$ denotes the maximum size of perturbations.

**Catastrophic Overfitting:** Wong et al. [2020] believe that non-zero initialization for perturbations is the key to avoiding the overfitting issue and propose to add uniform random noise during each training iteration. The detailed procedure is illustrated in the following equations:

$$
\begin{aligned}
\delta &= Uniform(-\epsilon, +\epsilon) \\
\delta &= \delta + \alpha \operatorname{sign}(\nabla_x \mathcal{L}(f_\theta(x), y)) \\
\delta &= \max(\min(\delta, \epsilon), -\epsilon)
\end{aligned}
\tag{4}
$$

Andriushchenko and Flammarion [2020] propose a regularization method GradAlign to maximize the gradient alignment between various sets as:

$$
\mathbb{E}_{(x,y) \sim D} \big[ 1 - cos(\nabla_x \mathcal{L}(f_\theta(x), y), \nabla_x \mathcal{L}(f_\theta(x + \eta), y)) \big]
\tag{5}
$$

where $\eta$ denotes random noise.

# 3   Bag of Tricks

We aim to investigate simple yet effective solutions to overcome the catastrophic overfitting in FGSM-AT. To stabilize FGSM-AT and make the trained model more robust to adversarial attacks, we propose strategies from three general perspectives: *Data Initialization, Network Structure, and Optimization*. In this section, the experiments are done on CIFAR-10 dataset Krizhevsky [2009] with PreActResNet-18 He et al. [2016] under the $\ell_\infty$ adversarial attack of maximal perturbation of $\epsilon = 8/255$ without using any additional data. Two kinds of adversarial attacks are designed to evaluate the robustness of models at the end of training: 10-steps projected gradient descent attack (PGD-10) Madry et al. [2018] and the standard version of AutoAttack (AA) Croce and Hein [2020b]. Specifically, for the PGD-10 attack, we apply untargeted mode using the ground-truth annotations with a step size $\alpha = 2/255$. AutoAttack comprises AutoPGD-CE, AutoPGD-Targeted, FAB Croce and Hein [2020a], and Square attack Andriushchenko et al. [2020].

**Default setting.** We set the training framework and hyper-parameters following Pang et al. [2021]. We apply SGD optimizer with a momentum of 0.9, weight decay of $5 \times 10^{-4}$, and an initial learning rate of 0.1. ReLU function (without applying label smoothing) is used as the default activation function. For the CIFAR dataset, we apply random flip and random crop as data augmentation methods. Following the framework settings in Pang et al. [2021], all models are trained for 110 epochs. The learning rate decays at $105^{th}$ and $110^{th}$ epochs. Specially, we report the robustness results on the last checkpoint. It should be noted that the final result might not be the best during the training process. Our experiments are conducted with NVIDIA TITAN XP GPUs.

| Methods | AT | PreActResNet-18 | | | WideResNet-34-10 | | |
|---|---|---|---|---|---|---|---|
| | | Clean | PGD-10 | AA | Clean | PGD-10 | AA |
| Baseline | F+FGSM | 86.4% | 46.7% | 41.0% | 89.4% | 0% | 0% |
| | FGSM+GradAlign | 81.2% | 48.7% | 44.0% | 81.2% | 48.7% | 44.0% |
| | PGD-10 | 82.6% | 53.1% | 48.3% | 86.1% | 56.5% | 52.2% |
| Data initialization | FGSM-Mask | 82.5% | 50.0% | 44.2% | 79.9% | 33.7% | 29.7% |
| | FGSM-Mask-fixed | 80.7% | 48.6% | 43.1% | 72.3% | 24.3% | 20.9% |
| Network Structure | FGSM-Smooth | 74.8% | 48.5% | 43.1% | 75.6% | 48.6% | 44.2% |
| | FGSM-Str2 | 83.1% | 48.7% | 44.4% | 85.0% | 50.4% | 46.7% |
| Optimization | FGSM+GradNorm | 82.4% | 47.2% | 42.7% | 82.8% | 50.7% | 46.2% |
| | FGSM+WeightNorm | 81.7% | 48.3% | 42.8% | 85.7% | 48.8% | 45.7% |

Table 1: Robustness performances of various methods on PreActResNet-18 and WideResNet-34-10.

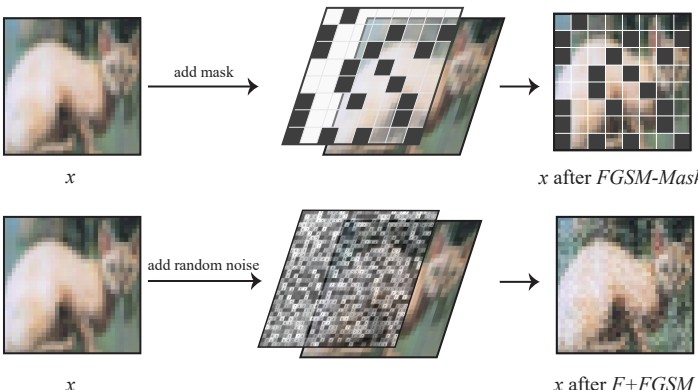

Figure 1: FGSM-Mask V.S. F+FGSM on an input image

## 3.1 Data Initialization

Wong et al. [2020] firstly identify the catastrophic overfitting faced in FGSM-AT and propose to resolve this issue by initializing images with uniform noise with size $\alpha = \epsilon$, namely Fast FGSM-AT (F+FGSM). As shown in Equation (4), the method is also termed "random initialization" since it randomly adds uniform perturbations during different training iterations. This method has been shown the capability to prevent general catastrophic overfitting and defend the models from PGD attacks.

**FGSM-Mask.** Inspired by the core idea of F+FGSM, in this paper, we propose to *mask* randomly a proportion of the input pixels to stabilize the training procedure of FGSM-AT, which we term as FGSM-Mask. Fig 1 demonstrates the comparison of FGSM-Mask and F+FGSM when generating adversarial examples. In each iteration, FGSM-Mask zeros out some randomly chosen pixels of each image $x$ with a mask $M$ according to a given mask ratio. Then the masked image $x \otimes M$ is fed to the model to generate adversarial examples via FGSM as:

$$\delta = \alpha \operatorname{sign}(\nabla_{x \otimes M} \mathcal{L}(f_\theta(x \otimes M), y)), \tag{6}$$

Compared with the random initialization method in F+FGSM (Equation (4)), our method exhibits a much simpler form—Our FGSM-Mask simply randomizes the *mask* instead of manipulating the original pixel values.

To demonstrate the effectiveness of our FGSM-Mask, we mask images with different ratios and present the robust accuracy in Table 2 and Figure 2 (a). With a mask ratio of 0%, our method is reduced to the vanilla FGSM-AT, and therefore it suffers from catastrophic overfitting. As the mask ratio increases, the models trained with FGSM-Mask become more stable. A small mask ratio like 10% or 20% can already attenuate the overfitting issue but the robust accuracy still drops to near-zero after decreasing the learning rate. With a mask ratio higher than 30%, the catastrophic overfitting is entirely resolved: the robust accuracy stably remains at 50.0%, outperforming F+FGSM (46.7%) by more than 3%.

**FGSM-Mask-Fixed.** Additionally, we observe that the randomness of masking is not necessary for different training iterations. Instead, simply using a fixed masking pattern throughout the training process is enough to help stabilize FGSM-AT. It is worth to be noted that the AT with fixed masks is equivalent to preparing a pre-defined masked adversarial dataset which will be fixed in the entire training process. The model trained with such a

| Randomized Mask Ratio | Robust Accuracy | Fixed Mask Ratio | Robust Accuracy |
|---|---|---|---|
| 0~20% | 0% | 0~20% | 0% |
| 30% | 50.0% | 30% | 0% |
| 40% | 49.3% | 40% | 48.6% |
| 50% | 49.0% | 50% | 48.6% |

Table 2: Robust accuracy V.S. mask ratio for FGSM-Mask and FGSM-Mask-Fixed.

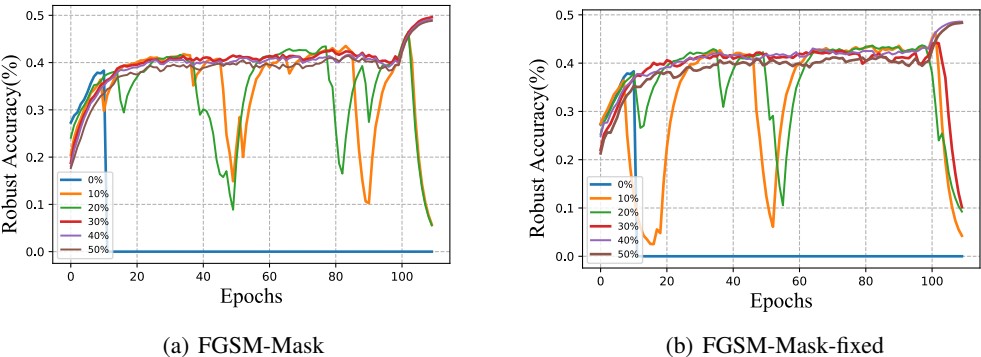

(a) FGSM-Mask

(b) FGSM-Mask-fixed

Figure 2: Robust accuracy of FGSM with various mask ratios. (a) is with the random mask, and (b) is with the fixed mask.

masked dataset achieves remarkably stable and decent robust accuracy without applying any additional tricks, as shown in Table 2. We call this method FGSM-Mask-Fixed. Similar to FGSM-Mask, with relatively lower mask ratios ($\geq 30\%$), the catastrophic overfitting cannot be fully resolved by FGSM-Mask-Fixed, and the trained model result in a final robust accuracy of 0%. As shown in Figure 2 (b), when increasing the mask ratio to 50%, the model trained with FGSM-Mask-Fixed reaches a robust accuracy of 48.6%, outperforming F+FGSM by about 2%. To show how randomized mask ratios and the fixed mask ratios influence the final robustness performance, Table 2 presents the robust accuracy with both FGSM-Mask and FGSM-Mask-Fixed under various mask ratios.

The findings in our *Data Initialization* section challenge the traditional belief that the randomness of initialization in different training iterations plays a crucial role in AT Chen et al. [2020], which inspires us to revisit the data initialization strategy in F+FGSM. We further find that it is not necessary to pursue the randomness of uniform noise during different training epochs. Instead, fixing the uniform noise of F+FGSM can also stabilize the FGSM-AT and finally reach a robust accuracy of 46.5% under PGD-10 adversarial attack, which is comparable to the vanilla F+FGSM.

## 3.2 Network Structure

Existing studies have demonstrated that a well-designed network structure can improve the model robustness. Xie et al. [2020], Singla et al. [2021], Wu et al. [2020]. When trained with FGSM-AT, Vision Transformers (ViTs) Dosovitskiy et al. [2020] have shown better robustness compared with CNNs Bai et al. [2021], Paul and Chen [2021], Shao et al. [2021]. Furthermore, Xie et al. [2020], Singla et al. [2021], Gowal et al. [2020] effectively boost the model robustness by replacing the original ReLU activation function with smoother ones. However, these approaches only focus on improving the model robustness in the general training process, but have overlooked the potential value of network structure for addressing the catastrophic overfitting in FGSM-AT. In this section, we investigate the role of network structure in FGSM-AT following the ideas of ViTs and smooth activation functions.

**Larger stride for the first convolution layer.** We first examine whether using ViTs can resolve the overfitting issue. We implement vanilla FGSM-AT with the compact Vision Transformer (CVT) Hassani et al. [2021], a Transformer architecture designed for the dataset with a smaller resolution. We observe that the robust accuracy under PGD attacks does not drop to zero during the whole training process, without applying any other tricks, neither random initialization nor regularization. The fact that ViTs can successfully avoid the overfitting issue motivates us to rethink whether we can achieve the same goal simply by modifying the architecture of CNNs. As one big difference between ViTs and CNNs lies in how they process images at the beginning of the network, we propose to simply modify the first convolution layer of CNNs to approach the similar behaviour of ViTs. ViTs begin with a patchify operation, which splits an image into a sequence of non-overlapping patches. Whereas

for CNNs, taking PreActResNet-18 as an example, the first layer is a $3 \times 3$ convolutional layer with stride 1, which results in overlapping sliding windows when computing the convolution features. To mimic the behaviour of ViTs, we propose to enlarge the stride size of CNNs to reduce the overlapped regions between adjacent sliding windows. By simply increasing the stride size from 1 to 2 or 3, the catastrophic overfitting problem is successfully addressed. As shown in Figure 3, when the stride is set to be 1, the robust accuracy quickly drops to zero. When the stride is set to be 2 or 3, the robust accuracy curve performs much more stable. Among different stride options in our study, we find that FGSM-AT with a stride as 2 achieves the highest robust accuracy. Therefore we adopt this setting in later experiments, namely FGSM-Str2.

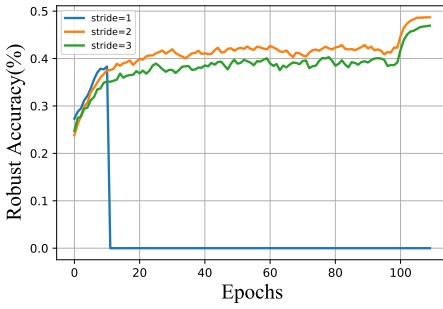

Figure 3: Robust accuracy and clean accuracy of Large Stride Size CNN. A larger stride size builds the robustness successfully.

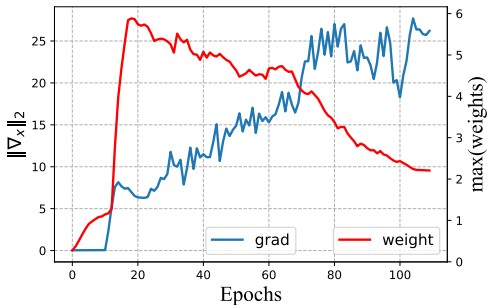

Figure 4: Trend of $\|\nabla_x\|_2$ and the maximum value of weights. Both increase dramatically when the overfitting happens.

**Smooth activation function.** We also investigate the role of the activation function in FGSM-AT. We replace the original ReLU activation function with smoother ones and then explore their effectiveness for addressing the overfitting problem. We select four smooth activation functions: SiLU Ramachandran et al. [2018], ELU Clevert et al. [2016], SoftPlus Nair and Hinton [2010], and GELU Hendrycks and Gimpel [2016]. We display the curves of these activation functions and record their robust accuracy during FGSM-AT in Figure 5(**a**). It can be observed that smooth activation functions can all mitigate or even fully prevent catastrophic overfitting.

We also find that the degree of smoothness affects the robustness. For instance, ELU is smoother than GELU and accordingly the robust accuracy of ELU is stabler than that of GELU. Following Xie et al. [2020], we choose SoftPlus to study the effect of function smoothness because the scaler $\alpha$ in Parametric SoftPlus can control its smoothness as the following:

$$f(\alpha, x) = \frac{1}{\alpha} \log(1 + \exp(\alpha x)). \tag{7}$$

Figure 5(**b**) shows the curves of SoftPlus when the $\alpha$ is 2, 5, 10 and the according robust accuracy curves. As $\alpha$ decreases, the activation function becomes smoother, and the robust accuracy becomes stabler. Figure 5(**b**) validates that the smoothness of activation functions has a positive correlation with the stability of FGSM-AT. Here we choose SoftPlus with $\alpha = 2$ as our baseline shown in Table 1 as it performs the best among smooth activation functions, and we call this method FGSM-Smooth.

### 3.3 Optimization

Adding an extra regularization term has been shown capable to prevent the catastrophic overfitting in FGSM-AT but can usually result in extra computation overhead. One typical example is GradAlign Andriushchenko and Flammarion [2020], which adds an additional objective to maximize the gradient alignment inside the perturbation set. GradAlign is quite effective for stabilizing FGSM-AT. However, it comes at the cost of an extra computational burden due to an extra forward and backward propagation to compute the gradient of an adversarial set $\nabla_x \mathcal{L}(f_\theta(x + \eta), y)$ (Equation (5)).

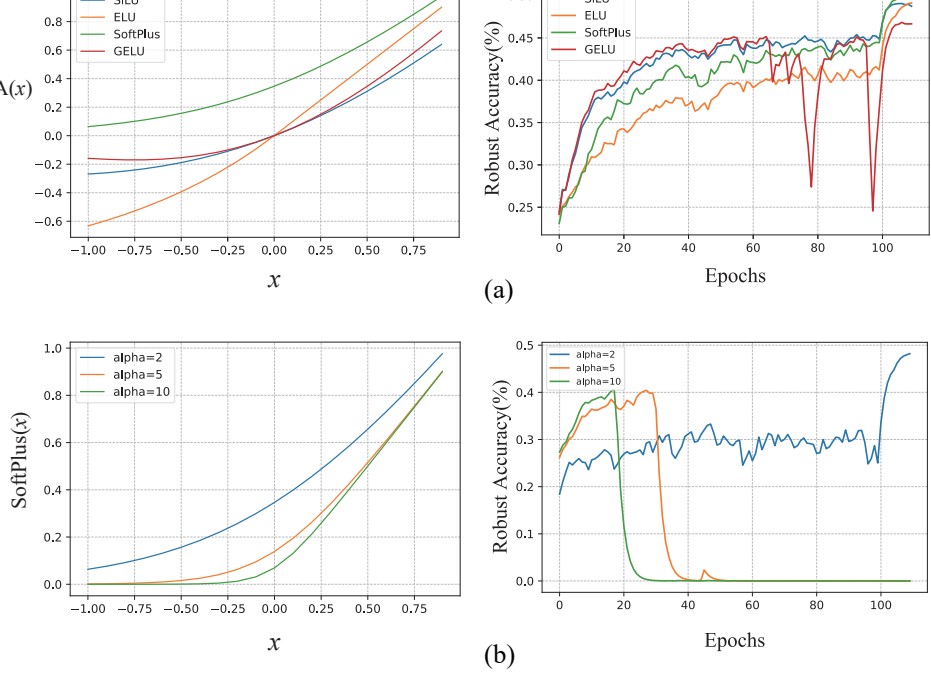

Figure 5: Curves of activation functions and their corresponding robust accuracy. (a) shows the comparisons between various activation functions. (b) shows the comparisons between Softplus with different $\alpha$.

In this paper, to avoid the extra forward propagation in GradAlign, we first introduce a novel regularization method which directly regularizes the $L_2$ norm of gradients on input images, referred to as *GradNorm*. Then to further reduce the computation cost, we design another simple but efficient method by only regularizing the weights on the first layer, referred to as *WeightNorm*. Both GradNorm and WeightNorm successfully address the overfitting issue and achieve comparable robust accuracy with GradAlign, while WeightNorm significantly reduces the computational cost. For instance, WightNorm and GradAlign take 42 seconds and 56 seconds for each training epoch. WeightNorm is 24% faster than GradAlgin. Next we illustrate the technical details of GradNorm and WeightNorm.

**GradNorm.** By taking a closer look at the $L_2$ norm of gradients on input images, we observe that the $\|\nabla_x\|_2$ becomes $100\times$ larger after the $11^{th}$ epoch as shown in Figure 4. This observation aligns with the conclusion in Kim et al. [2021], which points out that the increasing gradient norm leads to decision boundary distortion and a highly curved loss surface during adversarial training. This distortion hence makes the adversarially trained model vulnerable to multi-step adversarial attacks (*e.g.*, PGD attacks) and leads to catastrophic overfitting. This phenomenon inspires us to design a new regularizer by directly constraining the gradient norm $\mathbb{E}[\|\nabla_x\|_2]$:

$$\mathcal{L} = \mathcal{L}(f_\theta(x + \delta), y) + \beta\|\nabla_x\|_2 \tag{8}$$

where the hyper-parameter $\beta$ controls the weight of the regularizer. As shown in Table 1, GradNorm successfully overcome the overfitting issue and achieves a high robust accuracy of $47.2\%$ against PGD-10 attacks, which is comparable to the result of GradAlign ($48.7\%$).

**WeightNorm.** Both GradAlign and GradNorm are highly effective in addressing the overfitting issue. However, as aforementioned, they both suffer from high computational cost due to the additional back-propagation requirement. We hereby aim to design a novel regularization method which addresses the overfitting issue without introducing an extra computational burden. We propose *WeightNorm*, a regularization term that directly exploits the intermediate features of vanilla FGSM-AT models. Since the goal of adversarial training is to let the predictions of adversarial examples close to that of clean samples as much as possible: $f_\theta(x + \delta) \rightarrow f_\theta(x)$, we design to optimize the training process

by constraining the prediction difference. For simplicity, we only examine initial features generated by the first convolution layer $f_\omega$, where the $\omega$ denotes the weights of the first convolution layer. The $f_\omega(x + \delta) - f_\omega(x)$ can be represented as $\omega(x + \delta) - \omega x$, which is equal to $\omega\delta$. Therefore, therefore, pushing $f_1(x + \delta) \to f_1(x)$ is to minimize $\omega\delta$. We can either regularize the $\delta$ (*i.e.*, gradients of images) or the weights $\omega$.

Regularizing $\omega$ is cheaper than constraining the image gradient (which is essentially $\delta$) as only a part of model parameters are regularized, which avoids the second-order back propagation. After observing the change of $\omega$, we find that the maximum value of the weights also significantly increases when the catastrophic overfitting occurs. As shown in Figure 4, larger weights suggest that the network overfits the training data. Therefore, we design a regularizer aiming at constraining $\omega$. The intuition of this regularizer design is to both avoid large values in weights and also reduce the distance between clean features generated by clean samples and adversarial features generated by adversarial samples. We select $L_1$ norm to define the regularizer as:

$$\min_\omega \lambda\mathcal{L}^1(f_\omega(x), f_\omega(x + \delta)) \tag{9}$$

where the $\lambda$ controls the weight of the regularizer and $\delta$ is the adversarial perturbation. The proposed regularizer constrains $\omega$ and pushes the first-layer intermediate features of adversarial examples to be closer to that of clean samples. Experiments show that this regularizer could prevent the catastrophic overfitting and it does not require an extra forward pass like GradAlign shown in Equation 5.

### 3.4 Combination of Tricks

Each approach we propose can mitigate the catastrophic overfitting problem individually. To investigate the aggregated effect, we combine some of them and show results in Table 3. Adding the mask to images and increasing the stride size at the same time do not improve the performance. WeightNorm does not benefit other tricks. Smooth activation function can benefit masking image or a large stride size, showing improvement in the robustness performances. After trying different combinations, we find that combining a large stride size and smooth activation functions have the best performance.

| Methods | | | | Performances | | |
|---|---|---|---|---|---|---|
| Mask | Large Stride Size | Smooth Activation Function | WeightNorm | Clean | PGD-10 | AA |
| ✓ | ✓ | | | 82.5% | 49.4% | 45.1% |
| ✓ | | | ✓ | 81.1% | 51.2% | 46.1% |
| | ✓ | ✓ | | 82.2% | 51.3% | 46.4% |
| | ✓ | ✓ | ✓ | 81.3% | 51.2% | 46.1% |

Table 3: Performances of FGSM-AT with combined tricks

## 4 Scalability to Large Networks

Compared with small networks, the larger networks are more likely to overfit the training data as the network parameters increase, and the mentioned tricks might not work. As displayed in Table 1, when the size of the network increases (from PreActResNet-18 to WideResNet-34-10), F+FGSM results in 0% of robust accuracy under the adversarial attack. To comprehensively validate the effectiveness of the methods mentioned above, we conduct experiments on WideResNet-34-10 with the same training recipe as PreActResNet-18. Table 1 exhibit the robustness performances of different methods on these two networks, and the displayed results are taken at the final checkpoint. For the masking methods in *Data Initialization*, the mask ratio is set larger on WideResNet. Compared with PreActResNet, the effectiveness of masking methods declines, but they still exhibit higher robust performances than the vanilla F+FGSM. For the methods in *Network Structure*, both the FGSM-AT with a larger stride size (FGSM-Str2) and with smooth activation functions (FGSM-Smooth) perform stably on WideResNet, showing comparable results with PreActRest. On both PreActRestNet and WideRestNet, the FGSM-Str2 generally outperforms the other three tricks. Furthermore, the combination of tricks is also validated on WideRestNet. Following the optimal settings from Table 3,

we combine the smooth activation function and large stride size in FGSM-AT. With the combined tricks, the models respectively achieves a robust accuracy of 51.8% and 47.3% under PGD-10 attack and AA, outperforming all other FGSM-AT methods.

## 5 Related Work

**Adversarial training.** Adversarial training has been regarded as one of the most effective strategies to defend against the adversarial threats to machine learning systems. The idea of adversarial training origins in Goodfellow et al. [2015] who proposes to combine clean samples and adversarial examples to train the model. Madry et al. [2018] first demonstrate the optimization problem in adversarial training and proposes the PGD adversarial attack. Furthermore, advanced adversarial training methods are proposed. Zhang et al. [2019] apply a regularization term to achieve the balance between robustness and clean performance. Shafahi et al. [2019] reduce the high cost of adversarial training by recycling the gradient information. Carmon et al. [2019] first augment CIFAR-10 with 500K unlabeled extra data from 80 Million Tiny Images dataset. Some works also summarise the tricks of AT and the optimal settings for AT. Pang et al. [2021] list the optimal hyperparameters for PGD-AT on CIFAR-10 dataset. Gowal et al. [2020] introduce weight average(WA) to adversarial training and find the optimal ratios of extra data to get the best adversarial robustness.

**Catastrophic overfitting.** Though as an efficient method, FGSM-AT is not popular now because of its failure against severe attacks, like PGD adversarial attack. Wong et al. [2020] first find that the robust accuracy under PGD adversarial attack of FGSM-AT will drop to zero after several epochs, and this phenomena is named as catastrophic overfitting. Rice et al. [2020] think that catastrophic overfitting is a special case only existed in FGSM-AT and this overfitting phenomenon is due to a weaker adversarial attacker. Kim et al. [2021] visualize the decision boundary during adversarial training and find the decision boundary distortion is closely related to the catastrophic overfitting. They believe that the fixed distance from adversarial examples to clean images are the key causing the distortion and propose to apply various step sizes for each image.

**Data initialization.** Data initialization has been a common trick in adversarial training, where random noise is added to images before AT during each iteration. Madry et al. [2018] first add a random start for PGD-AT. Tramèr et al. [2018] first propose R+FGSM combining a Gaussian random initialization in a single-step AT. They add Gaussian random noise to images and do FGSM-AT with a step size of $\alpha = \epsilon/2$, which is not effective against PGD adversarial attack. Wong et al. [2020] believe that non-zero initialization for perturbations is the key to avoiding overfitting and propose adding uniform random noise to prevent overfitting.

**Regularization.** Wong et al. [2020] point out that early stopping is an effective method to get a robust model trained by FGSM-AT, but the robustness underperforms as the training epochs are inadequate. Andriushchenko and Flammarion [2020] demonstrate that the catastrophic overfitting is irrelevant to the sizes of networks. Instead, the local non-linearity was the true reason. To prevent overfitting, they propose a regularization method called GradAlign, which maximizes the gradient alignment between various set to stop the catastrophic overfitting.

## 6 Conclusion

This study proposes a range of tricks to address the catastrophic overfitting in FGSM-AT and comprehensively examine their effectiveness on networks with different scales. Our results show that the proposed tricks can be simple yet effective solutions to stabilize FGSM-AT at a minimal computational cost. We hope this study could contribute to the achievement of a fully stabilized FGSM-AT in the future.

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
