# OpenReview forum: "Bag of Tricks for FGSM Adversarial Training"
_NeurIPS.cc/2022/Conference — NeurIPS 2022 Submitted_

### Official Review · Reviewer_ASuB · 2022-06-30

**Rating:** 5
**Confidence:** 5
**Soundness:** 2 fair
**Presentation:** 2 fair
**Contribution:** 3 good

**Summary:**

The paper focuses on the problem of catastrophic forgetting overfitting in Adversarial training using FGSM. It investigates the data initialization, network structure, and optimization of CIFAR-10. It shows that
* applying masking pattern to the training set,
* smooth activations and using ViTs, and
* constraining first layer convolution weights in the convolution layers

stabilizes FGSM adversarial training.


**Questions:**

Other than the questions in the previous section, I had the following clarifications and suggestions:

- Did the authors also check the effect of batch-norm and its variants on FGSM-AT? It would also be interesting to see if the regularizers like mixup, cutmix, augmix etc. can further be used as tricks for FGSM-AT.

- The intuition behind the attenuation is unclear to me. Can the authors comment on the reason for attenuation at some point for the masks? Further, did the authors experiment with less than 10% masking?

- Which architecture was used for conducting architectures in Table 2 and Figure 2 a). Was the same architecture used for all experiments other Table 1? Did the authors check the observations for the other architecture and observe similar behavior?

**Limitations:**

While the paper mentions that limitations are highlighted in Section 6, the limitations were missing from this section. Therefore, I would suggest including a separate section highlighting the limitations and societal impact of the paper; this can also be included in the supplementary material, but I believe the authors can easily accommodate half a page in the current manuscript.

**Strengths And Weaknesses:**

I am well-familiar with the literature and read the full paper in detail. Accordingly, now I’ll describe the strengths and weaknesses of the paper in the order of originality and significance, quality of the paper, and clarity.

---

### Originality and Significance

#### Strengths

* The paper’s motivation is sound; it investigates the catastrophic forgetting overfitting from different perspectives, including data initialization, network structure, and optimization. While many previous works have understood this issue, the paper showcases that these simple tricks can help mitigate this problem to a certain extent.

* The tricks are novel for FGSM adversarial training, especially the results showing that a pre-defined masking pattern to the training, the effect of ViTs, and regularization terms are simple, easy to implement, and effective in isolation, as demonstrated by the results.

* The paper does a good job of positioning the proposed work. Furthermore, it does a good job in related work (section 5) explaining the existing works. I appreciate the paper using the optimal hyper-parameters from Pang et al. 2021. for their experiments.

#### Weaknesses

I do not have any significant concerns regarding the originality and contributions of the paper to the community. One minor suggestion is to include a discussion with Pang et al. 2021 when discussing smooth activation functions in Section 3.2, as the advantage of smooth activations was shown in their paper.

---
### Quality

The conducted experiments are interesting and valuable to show the performance gains by the proposed tricks for FGSM-AT. However, they were not exhaustive for me to evaluate the proposed tricks conclusively. Therefore, I have the following suggestions and questions that would be helpful to gain more insights and should improve the paper:

* The paper does not provide confidence intervals, making it hard to judge the improvement gains. For example, in Table 1, the weight norm seems to provide no benefit over the grad norm; it is also hard to see the differences between FGSM-Smooth and FGSM-Str without the confidence intervals. Similarly, in Table 3, given the minute differences between various combinations, it is challenging to judge the contributions of each of them individually.
* The evaluation protocol was also unclear, is the masking done only during training or even during the evaluation? In my opinion, it should be done only during training to avoid obfuscated gradients.
* Lastly, the effect of \lambda from eq. 9 was not highlighted in the paper. Other than mentioning the time taken per epoch in line 221, it would be beneficial to mention the total training time for both the methods up to convergence.

---
### Clarity

The paper was easy to read; however, I have various suggestions regarding the paper’s writing, as elaborated below.

* In Figure 5, I would suggest including Relu for a comparison. Further, I don’t think the left figures in Figure 5 are essential in the main paper. They can be moved to the appendix and the space can be utilized for important text like WideResNet results mentioned in Section 4.
* I would suggest repositioning section 4 and explaining the architectures in the experimental settings, followed by comparing the architectures' observations in the following sections.
* The images in Figure 1 can be positioned side-by-side horizontal instead of vertical.
* According to the NeurIPS template, the caption should have been on top of the table.
* The paper had an incorrect usage of \citep and \citet. For instance, lines 22, 26, 95, 98, etc., should have used \citep, but the paper uses \citet for all citations.

#### Other writing suggestions:
* Line 35: Extra space before the period
* Line 87: Extra space after f
* Line 248 - Refer to the experiments being talked about here.
* A period in equation 2.
* While the operators may seem obvious, for uninformed readers, it would be helpful to define $\Pi$ , $x_t$, $t$ in section 2.
* Missing comma after equation 5
* The paper should also mention the number of GPUs used for ease of reproducibility in Line 108
* The notation of FGSM-Str2 is not self-explanatory.
* Therefore, therefore -> Therefore in line 238
* Extra space after the period in line 277
* I suggest using the conference proceedings references for the papers; in many cases, it refers to the ArXiv version.

---

### Reproducibility
The authors provide the code to replicate the results in the supplementary material.

---

> ### Author Response · Authors · 2022-08-02
> **Author response(1/2)**
>
> We first thank the reviewer for the detailed comments and the appreciation of our work. Regarding the suggestions on writing/format/typos, we will fix all of them in the next version. We address other concerns as below:
>
> Q1: Discussion with [1] when discussing smooth activation functions.
> A1: Thanks for bringing up this related work! We will discuss it in the next version.
>
> Q2:  The contributions of each method and the confidence interval.
> A2: Thanks for your suggestion. The main goal of this paper is to discover a range of simple solutions/tricks from many different perspectives to address the catastrophic overfitting issue in adversarial training, therefore the comparison among these proposed methods is not highlighted. Regarding your concern about the confidence interval, we found the differences in final robustness from multiple runs are relatively small. We will update the next version accordingly for including these additional results/analyses.
>
>
> Q3: The use of masking during training or testing.
> A3: The mask is only used in training. We will make it clear in the next version.
>
> Q4: The effect of \lambda from eq. 9.
> A4:  The eq.9 is robust w.r.t. different \lambda. For example, by setting \lambda from 6 to 12, the method can all stop the catastrophic overfitting as shown in this figure(https://ibb.co/JjsGZjy). We will add this ablation in the next version.
>
> Q5: Add ReLU to Figure 5 as a comparison and move Figure 5 to the appendix.
> A5: Thanks. We will follow your suggestion on adding ReLU to Figure 5. While moving Figure 5 to the appendix is an option for us, we would prefer to still keep it in the main paper, as Figure 5 can help to reflect that we are the first work on revealing the value of smooth activation function in handling the catastrophic overfitting issue.
>
>
> Q6: Time of convergence.
> A6: The coverage time depends heavily on the learning rate scheduler. In our default setup, most settings typically take 20 epochs to converge, therefore the total time up to convergence is about 14 minutes. The training curves in Figure 2 and Figure 3 also describe the convergence process.
>
> Q7:  Effect of batch-norm and its variants on FGSM-AT.
> A7: Thanks for bringing up this interesting question. Following your suggestion, here are the results: 1) if we use the running status of BN to generate adversarial examples during training, the catastrophic overfitting can be mitigated to some extent. But it’s unstable compared to our method; 2) if alternatively using Instance Normalization, it cannot stop the catastrophic overfitting. We plan to try others like Layer Normalization and Group Normalization, and will update the results accordingly(https://ibb.co/djMh5px).
>
> Q8: Effect of data augmentation on FGSM-AT.
> A8: We tried several popular data augmentation methods, like Mixup and Cutmix, but they all fail to stop catastrophic overfitting.
>
> Q9: Which architecture was used for conducting architectures in Table 2 and Figure 2?
> A9: We use PreActNet-18 for all tables and figures except for the last three columns in Table 1 (which use WideResNet). We will make it clearer in the next version.
>
>
> Q10: The intuition behind the attenuation is unclear.
> A10: Firstly, we would like to stress that there are certain insights for motivating us to study these tricks empirically, e.g., adding masks provides another instantiation of applying randomness in FGSM-AT, GradNorm and WeightNorm are inspired from GradAlign. We will make it more clear in the next version.
>
> Secondly, rather than providing an in-depth analysis on explaining why one specific method works, this paper alternatively focuses on extensively showing the possibility of addressing the catastrophic overfitting problem from many different perspectives. In addition, we provide strong evidence to challenge the prior belief that randomness is the key factor in stabilizing adversarial training. For example, our methods (except FGSM-Mask) do not introduce any randomness into FGSM-AT but can still reliably secure model robustness. To our best knowledge, this is the first work of such type and we hope our work can provide a new perspective on motivating future work on comprehensively understanding and addressing catastrophic overfitting in adversarial training.
>
>
> Q11: Did the authors experiment with less than 10% masking?
> A11: Yes, the training curve here (https://ibb.co/VDYpfGN). We can observe that the robust accuracy drops to zero sometimes, especially when the learning rate decays.
>
> Q12: Reposition section 4 and explain the architectures in the experimental settings.
> A12: Thanks. We will reshape Section 4 following your suggestion.
>
>
> [1] Pang, T., Yang, X., Dong, Y., Su, H., & Zhu, J. (2021). Bag of Tricks for Adversarial Training. ArXiv, abs/2010.00467.
> [2] Andriushchenko, M., & Flammarion, N. (2020). Understanding and Improving Fast Adversarial Training. ArXiv, abs/2007.02617.

---

> > ### Author Response · Authors · 2022-08-08
> > **Author Response(2/2)**
> >
> > We run the experiments in Table 1 for three times with various seeds.
> > | Method(PreActResNet-18) | Clean | PGD-10-1 | AA |
> > | :-----| ----: | :----: |:----: |
> > |FGSM-Mask|82.4±0.01|49.6±0.01|44.2±0|
> > |FGSM-Mask-Fixed|80.7±0.01|48.5±0.02|43.0±0.02|
> > |FGSM-Smooth|74.9±0.09|48.2±0.07|43.0±0.03|
> > |FGSM-Str2|83.1±0.02|48.8±0.05|44.4±0.03|
> > |FGSM-GradNorm|82.3±0.03|47.3±0.04|42.7±0.04|
> > |FGSM-WeightNorm|81.7±0.02|48.4±0.00|42.8±0.01|
> >
> >
> > | Method(WRN-34-10) | Clean | PGD-10-1 | AA |
> > | :-----| ----: | :----: |:----: |
> > |FGSM-Mask|80.0±0.05|35.9±0.36|31.6±0.33|
> > |FGSM-Mask-Fixed|71.8±0.24|24.8±0.34|21.0±0.22|
> > |FGSM-Smooth|75.2±0.13|48.3±0.10|44.1±0.09|
> > |FGSM-Str2|85.1±0.13|49.6±0.33|46.0±0.31|
> > |FGSM-GradNorm|82.6±0.13|50.6±0.08|46.1±0.05|
> > |FGSM-WeightNorm|84.6±0.23|48.2±0.30|44.8±0.24|

---

> ### Author Response · Authors · 2022-08-09
> **Thanks again for your comments**
>
> Dear Reviewer ASuB
>
> Thanks again for your comments, which are quite helpful for us to improve the overall quality of this paper.
>
> Could you please let us know if our rebuttal addresses your concerns? If yes, are you willing to increase your score?
>
> Best regards
> Authors

---

> > ### Comment · Reviewer_ASuB · 2022-08-10
> > **Thank you for the response**
> >
> > I appreciate the author's response and the clarifications. However, I will keep my current score as I believe that the paper can be significantly improved by including the suggested tricks and lessons learned in the future version (I would also recommend the authors to read https://arxiv.org/abs/2010.03593), improving the clarity by justifying the proposed tricks for FGSM-AT, and including the relevant comparisons as mentioned by other reviewers. Therefore, I encourage the authors to address these comments and resubmit in the future.

---

### Official Review · Reviewer_zdGm · 2022-07-08

**Rating:** 6
**Confidence:** 4
**Soundness:** 3 good
**Presentation:** 3 good
**Contribution:** 3 good

**Summary:**

This paper investigates training tricks used for FGSM-AT, including data initialization, network structure and optimization. The combined tricks can effectively alleviate the catastrophic overfitting in FGSM-AT, making FGSM-AT a more practical way towards robustness.

**Questions:**

My main concern is that the proposed tricks largely hurt the clean accuracy.

**Strengths And Weaknesses:**

Strengths:
- The catastrophic overfitting in FGSM-AT is an important research topic.
- The investigated tricks like random masking / increasing stride size are interesting and underexplored in the literature.
- WeightNorm seems an efficient substitute for GradNorm.

Weaknesses:
- My main concern is that the proposed tricks largely hurt the clean accuracy (e.g., Table 1 FGSM-Mask drops clean accuracy from 89.4% to 79.9% on WRN-34-10; FGSM-Smooth drops clean accuracy from 86.4% to 74.8% on PreActResNet-18).

---

> ### Author Response · Authors · 2022-08-02
> **Author response**
>
> We first thank the reviewer for the detailed comments and the appreciation of our work. We address the concerns below:
>
> Q1: the proposed tricks largely hurt the clean accuracy.
> A1: We agree with the reviewer that, compared to our other methods, FGSM-Mask and FGSM-Smooth show a worse clean accuracy. Nonetheless, firstly, we would like to stress that the main point of this paper is to extensively show the possibility of addressing the catastrophic overfitting problem from many different perspectives; therefore as long as these methods successfully handle catastrophic overfitting with FGSM-AT, we believe they are legitimate solutions and worthy to be presented to the general community. Secondly, though exclusively applying FGSM-Mask or FGSM-Smooth hurts clean accuracy a lot, as discussed in Section 3.4, we are able to attain a reasonably high clean accuracy if combined FGSM-Mask or FGSM-Smooth with other methods. We will make these points clear in the next version

---

> > ### Comment · Reviewer_zdGm · 2022-08-04
> > **Thank you for the response**
> >
> > I totally agree that empirical evaluations like Pang et al. 2021 and this paper are valuable for the community. In FGSM-AT, the catastrophic overfitting problem is an important research topic, which is comprehensively discussed in this paper, so generally I vote for an acceptance. But I may not further improve my rating score since the drop on clean accuracy is a non-negligible drawback.

---

> > > ### Author Response · Authors · 2022-08-08
> > > **Thanks for positively supporting our work**
> > >
> > > Thanks for appreciating our work and being willing to vote for an acceptance.
> > >
> > > Yes, a standalone FGSM-Mask/Smooth hurts clean accuracy, and we agree with you that such an accuracy drop is non-negligible. While as shown in Table 3, a potential workaround is to combine FGSM-Mask/Smooth with our other methods, which can secure a reasonably high clean accuracy (and also a high robustness). More interestingly, though a standalone FGSM-Mask or FGSM-Smooth cannot work well, simply combining them together (i.e., FGSM-Mask-Smooth) successfully yields a model with good clean accuracy (81.1%) and robustness (46.1% against AA). Therefore, we believe that FGSM-Mask/Smooth may not be a major concern of this paper.
> > >
> > > Overall, thanks again for your comments. We will add this discussion to the next version for helping future readers better understand the values and also limitations of our work.

---

### Official Review · Reviewer_kmGb · 2022-07-09

**Rating:** 3
**Confidence:** 5
**Soundness:** 2 fair
**Presentation:** 3 good
**Contribution:** 2 fair

**Summary:**

This work investigates several modifications that can avoid Catastrophic Overfitting in FGSM-AT which would make AT much more efficient compared to using multi-step methods such as PGD. In particular, they suggest 4 different "tricks": Masking some pixels of the image, using a larger stride (default 2), using smooth activation functions (already suggested in prior work) and using a weight normalization regularizer. They empirically observe that some of their tricks achieve better performance than FGSM+GradAlign for CIFAR10 and $\epsilon=8/255$.

**Questions:**

1. Why do authors think that CVT does not suffer from CO? Have they tried larger perturbation radii?

2. It would be helpful to add a column for the computational cost (either in seconds per epoch or as a relative cost) for the different tricks and compared methods since one of the strengths of the proposed methods is efficiency.

**Limitations:**

As previously discussed, authors should test other datasets and larger perturbation radii to claim that the presented methods prevent CO.

**Strengths And Weaknesses:**

Strengths:

1. The topic of Catastrophic Overfitting and in particular efficient AT is relevant.
2. Results shown are encouraging.
3. Some results are surprising like the fact that a fixed mask works as well as randomly masking each image.

Weaknesses:

1. My main concern is that I do not think the experimental evidence is enough to support the claims. As reported in GradAlign work (https://arxiv.org/abs/2007.02617) some single-step AT methods (like F+FGSM) could prevent CO for moderate $\epsilon$ (such as 8/255) but would suffer from CO for larger radii. Therefore, the tested settings in this work (only CIFAR-10 and $\epsilon=8/255$) are too narrow to be reliable when stating that some particular method prevents CO. Additional perturbation radii should be compared and it would be sensible to include at least another dataset.

2. There is a lack of insight in the proposed tricks. While empirical results are definitely interesting, they should be discussed more in order to understand why do they work.

3. Some of the "tricks" are not very novel (e.g. smooth activation functions was already introduced in the context of adversarial training). Despite it is novel that they can mitigate CO in some settings, without more insight into why this would happen I consider the finding lacks impact.

---

> ### Author Response · Authors · 2022-08-02
> **Author response**
>
> We first thank the reviewer for the detailed comments. We address the concerns below:
>
> Q1:  the tested settings in this work are too narrow (only CIFAR-10 and $\epsilon$=8/255).
> A1: Thank you so much for these suggestions.
> We first test our methods on the popular CIFAR-100 dataset with $\epsilon$=8/255. As shown in the table below, while the vanilla FGSM fails to secure model robustness, all our methods can reliably address the catastrophic overfitting issue.
>
> | Method(CIFAR-100) |Clean | PGD-50-10|
> | :-----| ----:  |:----: |
> |Vanilla FGSM|46.7%|0%|
> |FGSM-Mask|56.4%|25.3%|
> |FGSM-Mask-Fixed|52.3%|22.4%|
> |FGSM-Smooth|51.6%|25.7%|
> |FGSM-Str2|57.9%|25.2%|
> |FGSM-GradNorm|51.6%|22.3%|
> |FGSM-WeightNorm|52.8%|23.1%|
>
>
> We then test our methods with a larger$\epsilon$=16/255 on CIFAR-10 and report the results below. We can observe that most of our methods can reliably address the catastrophic overfitting in the large perturbation radii setting.
>
> | Method |Clean | PGD-50-10|
> | :-----| ----: |:----: |
> |Vanilla FGSM|77.2%|0%|
> |FGSM-Mask|51.6%|20.4%|
> |FGSM-Mask-Fixed|50.1%|19.1%|
> |FGSM-Smooth|35.2%|25.2%|
> |FGSM-Str2|61.3%|0%|
> |FGSM-GradNorm|58.6%|25.1%|
> |FGSM-WeightNorm|59.3%|25.1%|
>
> We will add these additional results to the next version and hope they could address your concerns on insufficient empirical evaluations.
>
> Q2: a lack of insight into the proposed tricks.
> A2: Firstly, we would like to stress that there are certain insights for motivating us to study these tricks empirically, e.g., adding masks provides another instantiation of applying randomness in FGSM-AT, GradNorm and WeightNorm are inspired from GradAlign. We will make it more clear in the next version.
>
> Secondly, rather than providing an in-depth analysis on explaining why one specific method works, this paper alternatively focuses on extensively showing the possibility of addressing the catastrophic overfitting problem from many different perspectives. In addition, we provide strong evidence to challenge the prior belief that randomness is the key factor in stabilizing adversarial training. For example, our methods (except FGSM-Mask) do not introduce any randomness into FGSM-AT but can still reliably secure model robustness. To our best knowledge, this is the first work of such type and we hope our work can provide a new perspective on motivating future work on comprehensively understanding and addressing catastrophic overfitting in adversarial training.
>
> Q3: tricks (e.g., smooth activation functions) are not novel.
> A3: Firstly, we would like to stress that the main purpose of this paper is to offer a range of simple solutions/tricks from multiple perspectives to mitigate the catastrophic overfitting issue in adversarial training. We agree with the reviewer that some tricks were explored in adversarial training before, but, to our best knowledge, no prior works have (either empirically or theoretically) revealed their values in handling the important and challenging catastrophic overfitting problem. Therefore we believe our work is still novel and of interest to the general community.
>
> Q4: Why do authors think that CVT does not suffer from CO?
> A4: Our conjecture is mainly driven by recent works [1,2] on revealing the potential of vision transformers on adversarial robustness. We also confirm it empirically: by training CVT with a vanilla FGSM with $\epsilon$ on CIFAR-10, no catastrophic overfitting happens (see this training curve https://ibb.co/JKRYqSz).
>
> Next, following your suggestion, we try the large $\epsilon$=16. Nonetheless, we found CVT fails to learn anything meaningful during the training, i.e., clean accuracy and robustness always stay at 10% (i.e., random guess). Note this phenomenon is different from catastrophic overfitting (i.e., robustness first goes up and then is 0). We believe it relates to optimization difficulty (i.e., too many strong regularizations are added, including adversarial perturbations, mixup, cutmix, randaug) and we are still working on possible strategies (like warmup) to handle it. After tuning the strength of the regularizations, CVT can also become robust trained with adversarial examples with $\epsilon$=16(https://ibb.co/q9f6mXr). We will be keeping updating it during the discussion period.
>
> Q5: adding training time.
> A5: Please see the table below. In short, most of our methods do not bring in additional computation overheads to the vanilla FGSM-AT.
>
> | Method | Time(s/epoch) |
> | :-----|:----: |
> |Vanilla FGSM|40|
> |FGSM-Mask|40|
> |FGSM-Mask-Fixed|40|
> |FGSM-Smooth|43|
> |FGSM-Str2|37|
> |FGSM-GradNorm|109|
> |FGSM-WeightNorm|42|
>
> [1] Bai, Y., Mei, J., Yuille, A.L., & Xie, C. (2021). Are Transformers More Robust Than CNNs? NeurIPS.
> [2] Paul, S., & Chen, P. (2022). Vision Transformers are Robust Learners. AAAI.

---

> ### Author Response · Authors · 2022-08-08
> **follow up**
>
> Dear Reviewer kmGb
>
> Thanks again for your comments, which are quite helpful for us to improve the overall quality of this paper.
>
> Could you please let us know if our rebuttal addresses your concerns? Also, please feel free to let us know if you have more questions.
>
> Best regards
> Authors

---

> > ### Comment · Reviewer_kmGb · 2022-08-09
> > **Thank you for the effort in the rebuttal**
> >
> > Dear authors,
> >
> > After reading the authors' response and other reviews I am still convinced that this paper does not reach the bar for acceptance. Although I agree with other reviewers that the proposed tricks are novel and interesting. Everyone agrees that this work is purely empirical and as such, I think the experiments provided fall significantly short from previous literature:
> >
> > 1. **Several methods present CO for $\epsilon > 8/255$** One of the main observations from GradAlign [1] was that, although F+FGSM [2] or Free-AT [3] could prevent CO for moderate epsilons ($\epsilon=8/255$)  they would present CO for larger perturbation radii (see Fig 1 in GradAlign). *From GradAlign it is clear in the context of CO one should test their method against several epsilon radii while this work only uses $\epsilon=8/255$* -- I do not think additional experiments from the rebuttal are enough to address that.
> >
> > 2. **Missing relevant comparisons** In the setting of $\epsilon=8/255$ there are other *published works on single-step attacks that are not compared against* [4, 5]. Especifically note that to the best of my knowledge [5] is SOTA for single step attacks at  $\epsilon=8/255$. Additionally, there are also relevant preprints which have not been mentioned: [6, 7, 8, 9]. I think authors should acknowledge relevant ideas that have been proposed even if they do not compare against all of them.
> >
> > 3. **Overall narrow experimental setting** While I appreciate the authors performed ablations of their proposed methods, the overall experimental setting is too narrow (just CIFAR10 dataset and more importantly, just one $\epsilon$). All previous work on single-step attacks had performed evaluations on multiple datasets (usually one of them larger scale) and multiple perturbation radii.
> >
> > All in all, although I do think results presented look encouraging, the experiments performed are not enough to support the claim the proposed methods prevent CO and very relevant comparisons to published methods are missing.
> >
> > While I do not think this paper is ready to be published at this stage I do think there is potential for a relevant contribution to the community and would encourage the authors to re-submit this work after several key experiments have been added (see points above). I hope the reason for my score is clear and I am happy to provide clarifications for any of the points above.
> >
> > ---
> >
> > [1] Andriushchenko et. al. Understanding and Improving Fast Adversarial Training. NeurIPS 2020 - https://arxiv.org/abs/2007.02617
> > [2] Wong et. al.  Fast is better than free: Revisiting adversarial training. ICLR 2020 - https://arxiv.org/abs/2001.03994
> > [3] Shafahi et. al. Adversarial Training for Free! NeurIPS 2019 - https://arxiv.org/abs/1904.12843
> > [4] Sriramanan et. al. Guided Adversarial Attack for Evaluating and Enhancing Adversarial Defenses. NeurIPS 2020 - https://arxiv.org/abs/2011.14969
> > [5] Sriramanan et. al. Towards Efficient and Effective Adversarial Training NeurIPS 2021 - https://proceedings.neurips.cc/paper/2021/file/62889e73828c756c961c5a6d6c01a463-Paper.pdf
> > [6] Li et. al. Towards Understanding Fast Adversarial Training 2020 - https://arxiv.org/abs/2006.03089
> > [7] Golgooni et. al. ZeroGrad : Mitigating and Explaining Catastrophic Overfitting in FGSM Adversarial Training 2021 - https://arxiv.org/abs/2103.15476
> > [8]  Kang et. al. Understanding Catastrophic Overfitting in Adversarial Training 2021 - https://arxiv.org/abs/2105.02942
> > [9] de Jorge et. al. Make Some Noise: Reliable and Efficient Single-Step Adversarial Training 2022- https://arxiv.org/abs/2202.01181

---

> > > ### Author Response · Authors · 2022-08-09
> > > **Thanks for your comments**
> > >
> > > Dear Reviewer kmGb
> > >
> > > Thanks again for your comments. Regarding your concerns, we believe our rebuttal has reasonably addressed them. Specifically,
> > >
> > > **1. we additionally include the large $\epsilon=16$ experiments**, and found that most of our methods can reliably address the catastrophic overfitting in the large perturbation radii setting (see the 2nd table in our last response). We agree with you that the experiments could be more comprehensive by including more $\epsilon$ like {10, 12, 14}, while *given the observation in GradAlign that CO monotonically keeps getting severe by increasing $\epsilon$ (see Figure 1 in GradAlign paper)*, we expect our methods should secure robustness against other smaller $\epsilon$ like {10, 12, 14}. We will include these experiments in the next version.
> > >
> > > **2. we have compared with GradAlign** which is a strong baseline. More importantly, we would like to stress that the main point of this paper is to show that the pure FGSM-AT is capable of enabling robust learners (while prior works thought differently) and we leave achieving SOTA with FGSM-AT as future work.
> > >
> > > **3. we additionally include CIFAR-100 (see the 1st table in our last response)** and found that all our methods can reliably address the CO issue.
> > >
> > > In summary, we believe our paper (together with the additional experiments in the rebuttal) provides sufficient evidence on supporting the effectiveness of our methods in mitigating the CO issue in FGSM-AT. We hope this clarification can address your concerns.
> > >
> > > Thanks
> > > Authors

---

### Official Review · Reviewer_N6sV · 2022-07-10

**Rating:** 6
**Confidence:** 4
**Soundness:** 3 good
**Presentation:** 2 fair
**Contribution:** 3 good

**Summary:**

This paper proposes various methods to mitigate the drastic overfitting (and resulting worse robust test accuracy) that can occur when adversarial training using the fast gradient sign method (FGSM). The proposed methods include masking input pixels, increasing the stride size of the first convolutional layer, smoother activation functions, and regularization. The authors show that each of these approaches successfully mitigates catastrophic overfitting on its own, and that the combination of multiple of these techniques results in slightly better robust performance than any technique alone (including those in previous works).

**Questions:**

Questions
- Do you use a validation set or do you measure the robust accuracy using the test set during training?
- Did you re-train other methods in Table 1 or use downloaded weights?
- What is the main advantage of using the fixed mask, given it does not appear to outperform any instance of the random masking?

Suggestions
- It would be good to discuss the trade-off in clean performance amongst the different techniques. Additionally, it would be interesting to include clean performance in Table 2.
- You could better clarify what catastrophic overfitting is in the introduction for the reader that is unfamiliar with the term/phenomenon.

**Limitations:**

Yes

**Strengths And Weaknesses:**

Strengths
- This paper improves upon the performance of FGSM adversarial training by incorporating new techniques to mitigate catastrophic overfitting.
- The authors provide some light hypotheses behind why these techniques might work, and some empirical testing of these hypotheses.
- The paper is clearly written for the most part.

Weaknesses
- It is not made entirely clear which technique in the “bag of tricks” performs best and whether your combination of techniques is state-of-the-art (i.e. it would be good to state this clearly in the introduction, add bold font to your tables, and include the best of the results in Table 3 as a line in Table 1 as well).
- The paper could use another round of edits to improve readability: you should use \citep for parenthetical citations, and there are a few typos i.e. in lines 214-215.
- The experimental evaluation is not as thorough as in previous work - a stronger/more standard PGD evaluation uses 50 steps and 10 restarts rather than just 10 steps, and training time is excluded in the experimental results.
- While there are some attempts to explain why these techniques are successful at mitigating catastrophic overfitting, the paper could be strengthened by including only the most effective of the techniques, and providing a more thorough investigation into each technique.

---

> ### Author Response · Authors · 2022-08-02
> **Author response**
>
> We thank the reviewer for the detailed comments and the appreciation of our work. We address the concerns below:
>
> Q1: It’s not clear which method in “bag of tricks” performs best and whether the combination is SOTA.
> A1: Thanks! We will follow your suggestions to improve the overall presentation quality of this paper, including highlighting important points in the introduction and tables. In short, 1) all the tricks that we introduced can effectively prevent catastrophic overfitting in FGSM-AT, and, empirically, setting stride=2 and using a soft activation function can achieve the highest robustness (46.4% AA robustness); 2) our result is not SOTA on CIFAR-10; the main point of this paper is to show that the pure FGSM-AT is capable of enabling robust learners (while prior works thought differently) and we leave achieving SOTA with FGSM-AT as future work.
>
> Q2: another round of edits to improve readability, using \citep, and typos.
> A2: Thanks. We will further polish the paper and correct all format issues and typos.
>
>
> Q3: Evaluations against PGD-50 with 10 restarts and the training time.
> A3: Thanks for your suggestion. Please see the table below. We note 1) our methods all can reliably defend against PGD-50-10; and 2) the message delivered by PGD-50-10 is nearly the same as the message delivered by PGD-10-1, therefore the analysis/conclusions drawn in the original paper still hold.
>
> We additionally list the training time per epoch in the last column of the table. Compared to the vanilla FGSM, most of our methods do not introduce extra computational cost (i.e., the only exception is FGSM-GradNorm).
>
> We will add these results to the next version for completeness.
>
>
> | Method | PGD-50-10 | PGD-10-1 | Time(s/epoch) |
> | :-----| ----: | :----: |:----: |
> |Vanilla FGSM|0%|0%|40|
> |FGSM-Mask|48.7%|50.0%|40|
> |FGSM-Mask-Fixed|47.3%|48.6%|40|
> |FGSM-Smooth|47.0%|48.5%|43|
> |FGSM-Str2|47.2%|48.7%|37|
> |FGSM-GradNorm|45.9%|47.2%|109|
> |FGSM-WeightNorm|46.9%|48.3%|42|
>
> Q4: Use of the validation set or not during training.
> A4: We follow the default setup in [1,2], i.e., the robust accuracy is directly reported on the test set during training. We will make this setup clear in the next version.
>
> Q5: The main advantage of using the fixed mask, even if it does not appear to outperform any instance of the random masking.
> A5: Sorry for the confusion. The main point of using a fixed mask is to challenge the prior belief [2,3] that randomness (e.g., adding a Gaussian noise to the clean image as the initialization) is the key factor for stabilizing adversarial training. Our fixed mask experiment (at least empirically) supports that adversarial training can still be stable even without randomness. We hope this interesting phenomenon can shed light on future studies on stable adversarial training. We will make this point clear in the next version.
>
>
> Q6: Retrain other methods in Table 1 or use downloaded weights.
> A6: By building upon the public adversarial training repo https://github.com/P2333/Bag-of-Tricks-for-AT and https://github.com/tml-epfl/understanding-fast-adv-training, we train all methods from scratch. We will make this setup clear in the next version.
>
>
> Q7: Include only the most effective of the techniques, and provide a more thorough investigation into each technique.
> A7: Thanks for your suggestion. We would like to stress that our main point is to provide a set of “simple” solutions/tricks to alleviate the catastrophic overfitting issue in FGSM-AT. We hope our findings can encourage researchers to investigate model robustness from multiple perspectives and help lay the foundation of SOTA robust models in the future.
>
>
> Q8: Discuss the trade-off in clean performance amongst the different techniques.
> A8: Thanks for your suggestion. We will add such discussions to our paper. In short, we found that FGSM-Str2 + Smooth activation functions can reach the best balance between the clean accuracy and adversarial robustness.
>
>
> Q9: Clarify what catastrophic overfitting is in the introduction.
> A9: Thanks for your suggestion. We will revise the introduction accordingly.
>
>
>
> [1] Pang, T., Yang, X., Dong, Y., Su, H., & Zhu, J. (2021). Bag of Tricks for Adversarial Training. ArXiv, abs/2010.00467.
> [2] Wong, E., Rice, L., & Kolter, J.Z. (2020). Fast is better than free: Revisiting adversarial training. ArXiv, abs/2001.03994.
> [3] Madry, A., Makelov, A., Schmidt, L., Tsipras, D., & Vladu, A. (2017). Towards deep learning models resistant to adversarial attacks. arXiv preprint arXiv:1706.06083.

---

### Meta-Review · Area_Chair_HAH9 · 2022-08-20

**Recommendation:** Reject
**Confidence:** Certain

**Metareview:**

This paper proposes to solve FGSM catastrophic overfitting by combining different algorithmic methods (i.e., masking pattern to the train data, smooth activations, ViTs, constraints on the first layer convolutional weights).  The reviewers have considered the problem studied very relevant but were not convinced by the empirical evaluation, finding that the paper is missing an exhaustive evaluation (and for epsilon larger than 8). In addition, they would have appreciated some understandings on the different tricks considered. We encourage the authors to revise their paper, taking into consideration the reviewers’ feedback and to submit the revised work to a forthcoming conference.

**Award:**

No

---

### Decision · Program_Chairs · 2022-09-14

Reject